# Evaluation of Bronopol and Disulfiram as Potential *Candidatus* Liberibacter asiaticus Inosine 5′-Monophosphate Dehydrogenase Inhibitors by Using Molecular Docking and Enzyme Kinetic

**DOI:** 10.3390/molecules25102313

**Published:** 2020-05-14

**Authors:** Jing Nan, Shaoran Zhang, Ping Zhan, Ling Jiang

**Affiliations:** 1College of Horticulture and Forestry, Key Laboratory of Horticultural Plant Biology of Ministry of Education, Huazhong Agricultural University, Wuhan 430070, China; nanjing@webmail.hzau.edu.cn (J.N.); zhanping818@163.com (P.Z.); 2State Key Laboratory of Agricultural Microbiology, Huazhong Agricultural University, Wuhan 430070, China; zsr758991@163.com

**Keywords:** *Candidatus* Liberibacter asiaticus, Inosine 5′-monophosphate dehydrogenase, crystal, molecular docking, enzyme activity, antibacterial compound

## Abstract

Citrus huanglongbing (HLB) is a destructive disease that causes significant damage to many citrus producing areas worldwide. To date, no strategy against this disease has been established. Inosine 5′-monophosphate dehydrogenase (IMPDH) plays crucial roles in the de novo synthesis of guanine nucleotides. This enzyme is used as a potential target to treat bacterial infection. In this study, the crystal structure of a deletion mutant of *C*Las IMPDHΔ98-201 in the apo form was determined. Eight known bioactive compounds were used as ligands for molecular docking. The results showed that bronopol and disulfiram bound to *C*Las IMPDHΔ98-201 with high affinity. These compounds were tested for their inhibition against *C*Las IMPDHΔ98-201 activity. Bronopol and disulfiram showed high inhibition at nanomolar concentrations, and bronopol was found to be the most potent molecule (K_i_ = 234 nM). The K_i_ value of disulfiram was 616 nM. These results suggest that bronopol and disulfiram can be considered potential candidate agents for the development of *C*Las inhibitors.

## 1. Introduction

Huanglongbing (HLB) is one of the most destructive citrus diseases; it affects the citrus industry worldwide. HLB is a phloem-restricted, Gram-negative bacterium and caused by *Candidatus* Liberibacter asiaticus (*C*Las), *Candidatus* Liberibacter africanus, and *Candidatus* Liberibacter americanus. The pathogen is transmitted by the citrus psyllid [1]. *C*Las is highly virulent and distributed worldwide. HLB can infect all commercial varieties of cultivated citrus, and has caused enormous economic losses in the past [2]. Although some developments on *C*Las and plant-Liberibacter interaction have been achieved, no effective management method is presently available to control this disease once the trees are infected [3,4,5,6,7,8,9]. In September 2019, Ha and Beyenal from Washington State University were part of a research team that determined the process of culturing in a laboratory the bacteria that cause citrus greening. However, the culture technology of HLB has not been used in the development of drugs to control HLB [10]. Chemical control is considered to be an effective method to control citrus HLB. Controlling the transmit vector is a critical component which can slow down the spread, but it is not sufficient to eliminate this disease. Additional attempts have focused on the pathogen, and some broad-spectrum antimicrobials have been used against Ca. Liberibacter spp [11,12]. Streptomycin, penicillin G, and oxytetracycline reduce the titer of *C*Las in infected trees, but they also affect the native microbiota [13,14,15]. Compounds specifically targeting *C*Las have also been confirmed [16,17,18,19]. However, to date, no agents have been used commercially to combat this disease in the field. Targeting small-molecule inhibitors of pathogenic proteins is a new concept to control HLB. This process is potentially valuable to identify a new target to treat HLB infection.

Purine metabolism is critical for the growth and virulence of many bacterial pathogens [20,21,22,23]. Inosine 5′-monophosphate dehydrogenase (IMPDH) is the first and rate-limiting step in guanine nucleotide biosynthesis, controlling the gateway to guanine nucleotides. IMPDH catalyzes the oxidation of inosine 5′-monophosphate to xanthosine 5′-monophosphate (XMP) with a concomitant reduction of NAD^+^ to NADH. Guanosine 5′-monophosphate synthetase (GMPS) subsequently converts XMP to guanosine 5′-monophosphate (GMP). Almost every organism, except *Giardia lamblia* and *Trichomonas vaginalis*, has the IMPDH/GMPS pathway [24,25]. Bacteria can also acquire guanine nucleotides through the salvage pathways. However, in microbial infection, rapid proliferation places the demand upon the guanine nucleotide that the purine salvage pathway be insufficient for bacteria survival. The inhibition of IMPDH results in the depletion of guanine nucleotides. In recent decade, numerous IMPDH inhibitors have been used as anticancer, antiviral, and immunosuppressive agents [26,27,28]. IMPDH is also a promising target for antibacterial drug discovery [29,30,31,32].

The crystal structure of IMPDH in bacteria such as *Tritrichomonas foetus*, *Bacillus anthracis*, and *Pseudomonas aeruginosa* exists as a tetramer with a D4 square planar symmetry [33,34,35]. A monomer consists of two domains, namely, the catalytic domain, which is an eight-fold β/α barrel, and a subdomain, including two tandem cystathione-synthetase motifs (CBS domain or Bateman domain), which protrudes from the corners of the homotetramer [36]. The function of the CBS domain remains unclear. Deletion of the CBS subdomain by mutagenesis has little or no effect on enzymatic activity, but improves stabilization and crystallization [37,38,39].

To date, over 100 crystal structures of IMPDH have been added to the Protein Data Bank (PDB). Information on the binding sites between the protein and substrate, cofactor, or inhibitors is revealed from these crystal structures. Although eukaryotic and prokaryotic IMPDHs have similar overall structures, their kinetic properties and sensitivities to inhibitors are significantly different [40]. Structural comparisons revealed that the IMP binding site is well defined and highly conserved. By contrast, among IMPDHs, the cofactor site is more diverse, and species-specific inhibitors targeting this site have been developed [41]. One of the earliest reports discovered pathogenic IMPDH inhibitors in a high-throughput screening of small molecules against *Cryptosporidum parvum* IMPDH (*C*pIMPDH) [42]. Significant success has been achieved in the development of inhibitors of bacterial IMPDH, such as benzimidazoles, benzoxazoles, indazoles, triazoles, isobenzofurans, arylurea derivatives, and indoles [43,44,45,46,47,48,49,50,51,52,53,54,55,56,57]. Although bacterial IMPDHs have high sequence similarities and many species have the IMSM motif, the structure–activity relationships of each inhibitor are different [39,43]. Accordingly, simple predictions regarding compounds targeting a specific IMPDH are not sufficient, and experimental validation is necessary.

In this study, a *C*Las IMPDH variant without the CBS domain (*C*Las IMPDHΔ98-201) was designed. The recombinant *C*Las IMPDHΔ98-201 protein was expressed in the *Escherichia coli* system and purified using a Ni–NTA resin affinity chromatograph and high-resolution gel filtration column. The crystal structure of *C*Las IMPDHΔ98-201 was determined in the apo form. Repurposing drugs for a new target is increasingly used to find novel compounds. Eight known bioactive compounds were selected for molecular docking analysis, and the binding affinities were assessed. The inhibitions of these compounds against *C*Las IMPDHΔ98-201 activity were tested in vitro.

## 2. Results

### 2.1. Protein Purification of CLas IMPDHΔ98-201 and Crystallization Screening

To get the CBS deletion construct, 104 residues (98-201aa) were replaced with a G amino acid, and the catalytic residue Cys309 was completely conserved (Figure 1a). In the *E. coli* express system, recombinant *C*Las IMPDHΔ98-201 was soluble and stable. This protein was purified using a Ni-NTA resin affinity chromatograph and a high-resolution gel filtration column (Superdex 200), which showed a main peak (Figure 1b). *C*Las IMPDHΔ98-201 consisted of 390 amino acids with a theoretical molecular mass of 41 kDa. *C*Las IMPDHΔ98-201 appeared as a single band at approximately 40 kDa (Figure 1c).

The initial crystallization conditions that were tested from Index, SaltRx, PEG/Ion Screen, Crystal Screen kits (Hampton Research, Aliso Viejo, CA, USA), and the Wizard kit (Emerald BioSystems, Bainbridge Island, WA, USA). After the initial screening, crystals formed under two conditions only. After further optimization, diffraction-quality crystals were obtained by mixing 1 µL of protein solution at 8 mg/mL with 1 µL of reservoir solution (consisting of 30% PEG400, 200 mM sodium chloride, and 100 mM HEPES, pH 7.0) at 20 °C. Long rectangular crystals of approximately 0.2 × 0.1 × 0.05 mm formed (Figure 1d).

### 2.2. Crystal Structure and Loop Refinement of CLas IMPDHΔ98-201

Crystals of *C*Las IMPDHΔ98-201 appeared after 3 days at 293 K. The resolution of the diffracting crystal was 2.55 Å. Data collection and refinement parameters are summarized in Table 1.

The structure of *C*Las IMPDHΔ98-201 was determined through molecular replacement using IMPDH from *Campylobacter jejuni* (PDB entry 4R7J) as a template. Finally, the structure was refined to 2.55 Å resolution by using the PHENIX software. This crystal protein existed as a homotetramer (Figure 2a), which is well conserved in other IMPDHs. The space group of *C*Las IMPDHΔ98-201 was C121, and the unit-cell parameters for *C*Las IMPDHΔ98-201 were a = 143.13, b = 134.86, and c = 85.62 Å.

The structure of *C*Las IMPDHΔ98-201 was very well defined, and the refinement parameters were R_work_ = 22.2% and R_free_ = 26.5%. A comparison of the structures of *C*Las IMPDHΔ98-201 and BaIMPDHΔ95-200 (Figure 2b) showed that the structure of *C*Las IMPDHΔ98-201 was highly similar to that of *B*aIMPDHΔ95-200 in its apo form. The RMSD was 0.929 Å.

The flap loop and a C-terminal loop are not visible in the electron density. Thus, the loop refinement of *C*Las IMPDHΔ98-201 (PDB ID: 6KCF) was performed using Modeller 9.23 (Figure 2c). The nonterminal missing structure was refined (Figure 2d). Verification of the 3D results showed that 88.27% of the amino acid residues had an average 3D–1D score ≥ 0.2 (Appendix A). The Ramachandran plot analysis indicated that 82.4% of the residues were in the core region, 13.4% of the residues were in the allowed region, 2.9% of the residues were in the generously allowed region, and 1.3% of the residues were in the disallowed region (Appendix A).

### 2.3. Molecular Docking

The eight candidate compounds and the refined structure were selected to perform molecular docking. CDOCKER was used to perform a docking study of the selected molecule; the molecular docking binding affinities are shown in Table 2. Three molecules, namely, bronopol, mercaptopurine, and disulfiram, showed the -CDOCKER_ENERGY ≥ 10. Because mercaptopurine is an analog of IMP, it was hypothesized that bronopol and disulfiram would exhibit the best inhibitory effect for CLas IMPDH. The pose with the lowest binding energy was recognized as the most stable conformation for further structural analysis.

The 3D and 2D structures of the *C*Las IMPDHΔ98-201 with bronopol and disulfiram are displayed in Figure 3. Nine hydrogen bonds formed between bronopol and the residues ILE189, Gly190, Gly192, ASP228, Gly229, Gly230, Gly251, and Ser252 of *C*Las IMPDHΔ98-201 (Figure 3a,b). Disulfiram formed two hydrogen bonds with *C*Las IMPDHΔ98-201, namely, Ala41 and Ala42; four alkyl hydrophobic interaction with Met43, Pro190, and Met249; and one sulfur-x interaction with Met43 (Figure 3c,d). The 3D and 2D structures of the *C*Las IMPDHΔ98-201 with the rest of molecules are displayed in Appendix A.

### 2.4. Kinetic Characterization of CLas IMPDHΔ98-201

According to the standard assay conditions, the kinetic properties of *C*Las IMPDHΔ98-201 were as follows: K_cat_ = 7.2 ± 0.2 s^−1^; $K_{M}^{IMP}$ = 181 ± 19 µM (Figure 4a); and $K_{M}^{NAD+}$ = 318 ± 24 µM (Figure 4b). Similar to other IMPDHs, substrate inhibition was also observed at high NAD^+^ levels, $K_{ii}^{NAD+}$ = 7.3 ± 1.1 mM.

The steady-state parameters from other bacterial species are listed in Appendix A. All IMPDHs had similar K_m_ values for the substrate, but for *C*Las IMPDHΔ98-201, $K_{M}^{IMP}$ was the largest, and $K_{M}^{NAD+}$ was the smallest. These results indicate that *C*Las IMPDHΔ98-201 bound to IMP with the lowest affinity, but was the highest affinity binding NAD^+^ among the tested IMPDHs. The K_cat_ value may be due to the fact that the results described here were measured at 30 °C, whereas the other IMPDHs were measured at the lower temperature of 25 °C.

### 2.5. Inhibitory Assay against CLas IMPDHΔ98-201 Enzyme Activity

Extending the measurement time, no exponential enzyme decay against *C*Las IMPDHΔ98-201 was observed. Hence, the inhibition of bronopol, disulfiram, and ebselen was treated as a reversible mode (Appendix A). As shown in Appendix A, the V_max_ was found to be reduced with an increase in the inhibitor concentration, suggesting that bronopol inhibited *C*Las IMPDHΔ98-201 in a noncompetitive manner against IMP. Disulfiram also inhibited *C*Las IMPDH in a noncompetitive manner against IMP, where regression lines meet on the *X*-axis (Appendix A). The various types of inhibition by other small molecule inhibitors are summarized in Appendix A.

To study the mechanism of enzyme inhibition, the inhibition constant K_i_ with respect to the IMP substrates was measured at a fixed NAD^+^ concentration. The K_i_ values of these eight compounds are summarized in Table 3.

All values ranged from 0.234 µM to 3500 µM. Although the percentage of DMSO and the high concentration of the compound affected the stability of the target protein, the values for mizoribine and ribavirin may have been inaccurate (Appendix A). Ribavirin is a guanosine analog with broad-spectrum activity against RNA virus [58], and has almost no effect on the *C*Las IMPDHΔ98-201 enzyme activity. Mizoribine is an imidazole nucleoside which is used as an immunosuppressive agent [59]. Mizoribine was a potent inhibitor of IMPDHs, with K_i_ = 307.7 µM for *C*Las IMPDHΔ98-210, whereas the K_i_ value of *E. coli* IMPDH was 0.5 µM. Mercaptopurine yielded uncompetitive inhibition with K_i_ = 165 µM (Appendix A). Mycophenolic acid was shown to be a potent inhibitor of mammalian IMPDHs with K_i_ = 2.43 µM (Appendix A). Mycophenolate mofetil is a prodrug of mycophenolic acid [60], yielding K_i_ = 24.42 µM (Appendix A). Three compounds, namely, disulfiram, bronopol, and ebselen, have been repurposed as IMPDH inhibitors [61]. Bronopol had the best inhibitory effect with K_i_ = 234 nM (Figure 5a). The K_i_ value of disulfiram was 616 nM (Figure 5b). The K_i_ values of ebselen was 4.13 µM (Appendix A).

## 3. Discussion

*C*Las causes HLB and affects citrus. Although HLB has become a global problem, no effective HLB management strategy is available [11]. IMPDH is a validated target for the design of potent antibacterial agents, and the inhibition of this enzyme depletes cellular guanine nucleotides [36]. The development of inhibitors against bacterial IMPDHs has attracted increasing attention [62]. This study focused on the development of *C*Las IMPDH inhibitors. The first structure of *C*Las IMPDHΔ98-201 was determined. On the basis of its crystal structure, the refined structure was constructed, and molecular docking was performed to predict the binding energy. Then, we used an inhibition assay against *C*Las IMPDHΔ98-201 to validate the molecular docking predictions.

### 3.1. Purification and Crystallization of CLas IMPDHΔ98-201

To overcome the instability of *C*Las IMPDH, *C*Las IMPDH mutation was designed and purified. In the solution, recombinant *C*Las IMPDHΔ98-201 was more stable than the wild type. MtbIMPDH2 without the CBS domain displayed higher solubility [51]. The steady-state kinetics parameters of *C*Las IMPDHΔ98-201 were similar to those of other IMPDHs (Appendix A), suggesting that deleting the CBS domain would not affect the *C*Las IMPDHΔ98-201 catalytic properties [7]. Crystals of the apo form of *C*Las IMPDHΔ98-201 were obtained in 100 mM HEPES (pH 7.5) and 200 mM NaCl, with 30% (w/v) PEG 4000 as the precipitant. The RMSD of *C*Las IMPDHΔ98-201 and BaIMPDHΔ95-200 was 0.929 Å.

### 3.2. Docking Interaction Analysis of CLas IMPDHΔ98-201 with Molecules

To find inhibitors of *C*Las IMPDHΔ98-201, molecular docking was performed using Discovery Studio 2018. The docking scores of bronopol and disulfiram binding to *C*Las IMPDHΔ98-201 were −11.19 and −25.03 kcal/mol, respectively. Bronopol was stabilized by nine hydrogen bond interactions with residues ILE189, Gly190, Gly192, ASP228, Gly229, Gly230, Gly251, and Ser252. Additionally, disulfiram was stabilized by hydrophobic and sulfur-x interactions. Given that a flap loop and a C-terminal loop were not visible in the apo form structure of *C*Las IMPDHΔ98-201, the nonterminal missing structure of *C*Las IMPDHΔ98-201 was refined by Modeller. The Ramachandran plot and Verify 3D analysis suggested that the refined *C*Las IMPDHΔ98-201 structure was reliable. Bacterial IMPDHs were similar in sequence and structure (Appendix A). Homology modeling and in silico docking were performed to study the structure–activity relationship of indole derivatives against *Helicobacter pylori* IMPDH [63]. The crystal structure of IMPDH from *Cricetulus griseus* was prepared by using Discovery Studio 2.5 to build a pharmacophore model of IMPDH inhibitors and for the in silico docking analysis [64]. These studies supported the feasibility of molecular docking.

### 3.3. Inhibitory Assay against CLas IMPDHΔ98-201 Activity

To explore the inhibition of the eight compounds, an inhibitory assay against *C*Las IMPDHΔ98-201 activity was measured by monitoring the production of NADH. The inhibitions of *B*aIMPDH92-220, *C*jIMPDHΔ92-195, and *C*lpIMPDHΔ89-215 to a given compound showed significant differences, although the same residues interacted with the inhibitor [7]. A previous study found that a single residue showed mycophenolic acid resistance, although the binding sites were identical [65]. The kinetic mechanism was controlled for the mycophenolic acid resistance of *Pb*IMPDH-A and *Pb*IMPDH-B [66]. These studies showed that virtual screening by simple prediction is fast and low cost, although experimental verification is needed. Many other compounds against HLB have been reported. Five compounds, namely, C16, C17, C18, C19, and C20, were identified against *C*Las SecA, with IC50 values of 0.25, 0.92, 0.48, 0.64, and 0.44 µM, respectively [16]. ZINC05491830 is one of the most potent inhibitors of *C*Las Esbp, with an IC50 value of 2.59 µM [19]. ChemDiv C549-0604 is an inhibitors of *C*Las VisNR, with an IC50 value of 0.7 µM [18]. The inhibition assay suggested that the K_i_ values of bronopol and disulfiram were 234 and 616 nM, respectively. The inhibition of *C*Las IMPDHΔ98-201 suggested that bronopol and disulfiram, unlike the aforementioned other compounds, could be used as *C*Las IMPDHΔ98-201 inhibitors against other *C*Las genes.

## 4. Materials and Methods

### 4.1. Cloning and Mutant Construction of CLas IMPDH Gene

The coding sequence of IMPDH was amplified by PCR from the chromosomal DNA of *C*Las (strain psy62). The PCR product was cloned into the pET28at-plus expression vector.

The CBS domain deletion mutant (*C*Las IMPDHΔ98-201) was constructed via splicing overlapping extension polymerase chain reaction (PCR). The ΔS construct involved the deletion of 104 residues from M98 to T201. The *C*Las IMPDH gene in vector pET28at-plus was used as a template. The F1 and R1 primers were applied to amplify a region of *C*Las IMPDH ranging from residue M1 to residue M98. The F2 and R2 primers were used to amplify a region of *C*Las IMPDH ranging from residue T201 to residue I493. Codons for residues M98–T201 were replaced with codons for G. I1 and I2 were used as templates. PCR was performed to amplify the *C*Las IMPDH CBS domain deletion mutant gene by using the F1 and R2 primers. The *C*Las IMPDHΔ98-201 gene was digested by *Bam* HI and *Xho* I and inserted into a pET28a-SUMO vector. Then, pET28a-SUMO-*C*Las IMPDHΔ98-201 was transformed into *E. coli* BL21(DE3) cells.

### 4.2. Protein Purification and Crystallization of CLas IMPDHΔ98-201

Cells carrying pET28a-SUMO-*C*Las IMPDHΔ98-201 plasmid were cultured in LB media supplemented with 50 µg/mL of kanamycin at 37 °C. The culture was induced by adding 0.3 mM of isopropyl-β-D-thiogalactopyranoside when its OD_600_ reached 0.8–1.0. After 20 h of incubation at 16 °C, the cells were harvested by centrifugation at 6000 rpm for 6 min at 4 °C, resuspended in lysis buffer [20 mM Tris-HCl (pH 8.0), 500 mM KCl, 40 mM imidazole, 1 mM PMSF, and 10% glycerol], and then sonicated. The lysate was clarified by centrifugation at 16,000 rpm for 50 min at 4 °C. Clarified lysate was subsequently purified on a Ni–NTA agarose column, and the protein was eluted with the same buffer containing 500 mM imidazole. The SUMO tag was subsequently removed with the Ulp1 protease at 16 °C for 1 h. The target protein was additionally purified using a Ni affinity chromatograph to remove the released tag and uncut protein, followed by a size exclusion chromatography step on a Superdex^TM^ 200 (GE Healthcare) column equilibrated with buffer [20 mM Tris-HCl (pH 8.0), 100 mM KCl, and 10% glycerol]. All proteins were purified according to this protocol.

Crystallization screening was set up using the sitting drop vapor diffusion method in 96-well plates. Crystals of the protein appeared after 3 days at 293 K. The best crystals of *C*Las IMPDHΔ98-201 were obtained by mixing 1 µL of protein solution at 8 mg/mL with 1 µL of reservoir solution consisting of 30% (v/v) PEG 400, 100 mM HEPES (pH 7.5), and 200 mM sodium chloride.

### 4.3. Data Collection and Processing

Crystals were mounted on nylon loops and flash-cooled in liquid nitrogen. Diffraction data were collected at 100 K by using the Q315r CCD detector at the BL17U beamline of the Shanghai Synchrotron Radiation Facility. Single wavelength data at 0.9792 Å were obtained, and all data were processed and scaled with HKL3000 [67]. The structure of the *C*Las IMPDHΔ98-201 was solved by molecular replacement using PHENIX [68]. The refined model and structure factors were deposited in the PDB.

### 4.4. Loop Refinement and Molecular Docking

The 3D structure of *C*Las IMPDHΔ98-201 was used for refinement. The Modeller program was used to refine the nonterminal missing structure [69]. This refined structure consisted of 358 amino acids (*C*Las IMPDH 12-98 and 202-472). The PROCHECK validation server was used to check the quality of the refined model [70]. This structure was also validated by Verify 3D [71].

Molecular docking was performed using CDOCKER, a frequently applied module of Discovery Studio 2018. CDOCKER employs a CHARMm force field to calculate the binding free energy of the ligand to the receptor [72]. The eight filtered molecules used for docking were bronopol, ebselen, mercaptopurine, mizoribine, mycophenolate_mofetil, mycophenolic acid, ribavirin, and disulfiram. In the docking experiment, the refined structure of *C*Las IMPDHΔ98-201 was used as the receptor; the docking parameters are listed in Appendix A. The best pose of each molecular binding with a refined structure was estimated according to the binding energy. Interactions between the compound and protein, such as van der Waals force, hydrogen bond, electrostatic, hydrophobic, and halogen, were analyzed.

### 4.5. Steady-State Kinetics

Standard enzyme activity assay was performed in an assay buffer (50 mM Tris-HCl, 100 mM KCl, 1 mM DTT, pH 8.0) and a final *C*Las IMPDHΔ98-201 enzyme concentration of 100 nM at 30 °C. The production of NADH was monitored by the increase in absorbance at 340 nm (Ɛ = 6.22 mM^−1^ × 0007 cm^−1^). Apparent steady-state kinetic parameters were evaluated at varying concentrations of IMP (0.005–1 mM) and a fixed saturating concentration of NAD^+^ (3 mM), or at varying concentrations of NAD^+^ (0.005–5 mM) and a fixed saturation level of IMP (1 mM). Assays were performed in duplicate. The IMPDH enzymes displayed strong substrate inhibition with respect to NAD^+^ under the standard assay conditions. The method described by Kerr et al. was used to determine the kinetic constants [73].

### 4.6. Inhibition Assay against IMPDHΔ98-201 of CLas

The eight molecules that were purchased were screened in vitro. The assay was performed in a 200 µL final volume in a 96-well plate with a reaction buffer consisting of 50 mM Tris–HCl (pH 8.0), 100 mM KCl, and 1 mM dithiothreitol. Assays were performed using 100 nM CLas IMPDHΔ98-201 in the presence or absence of test compounds. The assay was allowed to proceed at 30 °C for 60 min.

The value of K_i_ for eight molecules was determined at a fixed saturation concentration of NAD^+^ (1 mM), different concentrations of IMP (0.02, 0.04, 0.08, 0.15, 0.25, and 0.5 mM), and in the presence of increasing concentrations of inhibitor. The concentrations of bronopol and disulfiram were 0.1, 0.2, 0.3, 0.4 µM. The concentration of ebselen was 1, 2, 3 and 4 µM. The concentration of mycophenolic acid was 1, 2, 4 and 8 µM. The concentration of mycophenolate mofetil ranged from 5 to 20 µM. The concentration of mercaptopurine ranged from 50 to 200 µM. The concentration of mizoribine was 500 and 750 µM. The concentration of ribavirin was 500 and 800 µM. Each determination of K_i_ was derived from duplicate measurements.

To determine the K_i_ values, the initial rate data versus substrate concentration at different inhibitor concentrations was fitted using Prism software (GraphPad) to equations for competitive, noncompetitive, or uncompetitive inhibition.

## 5. Conclusions

In conclusion, bronopol and disulfiram were confirmed as *C*Las IMPDH inhibitors. These results indicate that these compounds could be used as the lead scaffold to further design and develop potent *C*Las IMPDH inhibitors. However, the effect of compounds with activity against *C*Las was not tested. In future studies, we will focus on the effect of compounds in the treatment of HLB diseases. The apo form structure of *C*Las IMPDHΔ98-201 was solved, providing a means to study the complex structure of cocrystallization with inhibitors. The binding information may be helpful for the further development of antimicrobial compounds against *C*Las.

## Figures and Tables

**Figure 1 molecules-25-02313-f001:**
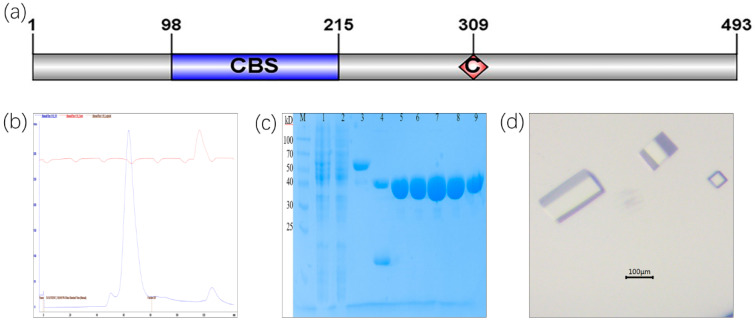
Size exclusion chromatogram and crystal of purified *C*Las IMPDHΔ98-201. (**a**): Primary sequence of *C*Las IMPDH; (**b**): Size exclusion chromatogram of *C*Las IMPDHΔ98-201; (**c**): SDS-PAGE of *C*Las IMPDHΔ98-201. M: protein marker; 1: supernatant; 2: flow-through; 3: SUMO-*C*Las IMPDHΔ98-201; 4: Ulp1 digestion; 5–9: protein of *C*Las IMPDHΔ98-201 after size exclusion chromatogram; (**d**): crystal of *C*Las IMPDHΔ98-201.

**Figure 2 molecules-25-02313-f002:**
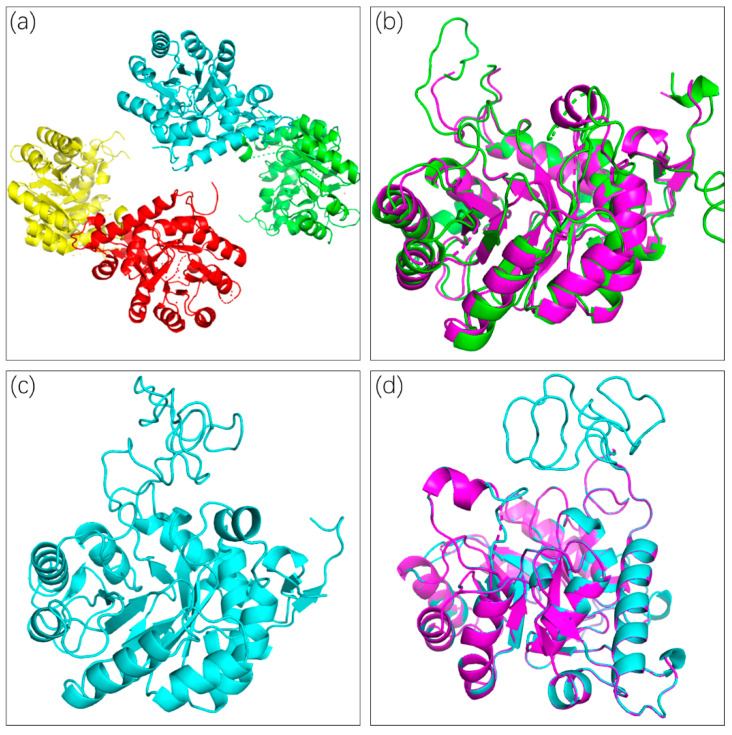
Crystal and Loop Refinement structure of the apo-form *C*Las IMPDHΔ98-201. (**a**): Tetramer of *C*Las IMPDHΔ98-201; (**b**): Superposed structures of *C*Las IMPDHΔ98-201 (PDB entry: 6KCF, in magenta) and *B*aIMPDHΔ95-200 (PDB entry: 4MJM, in green); (**c**): Loop refinement of *C*Las IMPDHΔ98-201; (**d**): Superposed structures of *C*Las IMPDHΔ98-201 (PDB entry: 6KCF, in green) and the refined structure (cyan).

**Figure 3 molecules-25-02313-f003:**
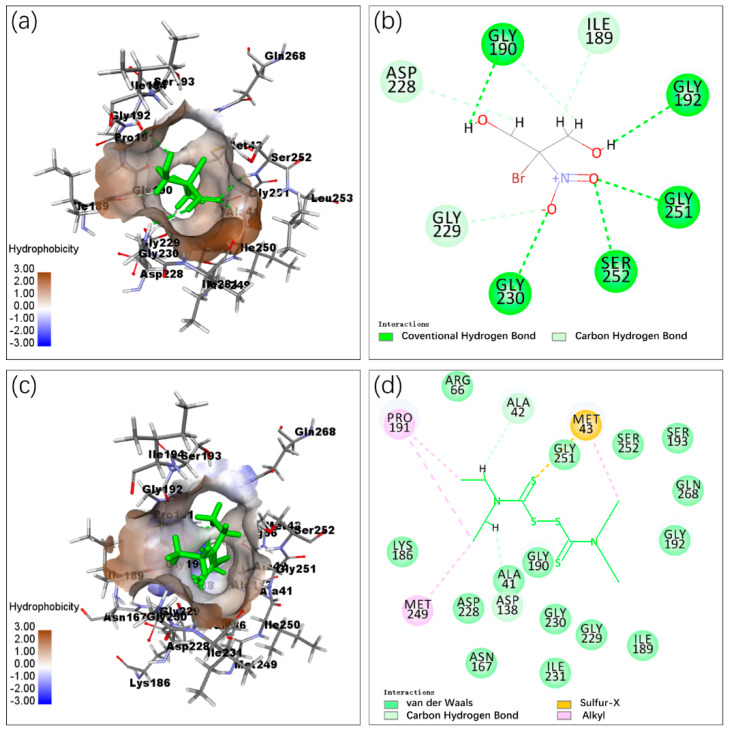
Molecular docking of *C*Las IMPDHΔ98-201 and the moleculars. (**a**): 3D details of *C*Las IMPDHΔ98-201 and bronopol (green) interaction; (**b**): 2D details of *C*Las IMPDHΔ98-201 and bronopol interaction; (**c**): 3D details of *C*Las IMPDHΔ98-201 and disulfiram (green) interaction; (**d**): 2D details of *C*Las IMPDHΔ98-201 and disulfiram interaction.

**Figure 4 molecules-25-02313-f004:**
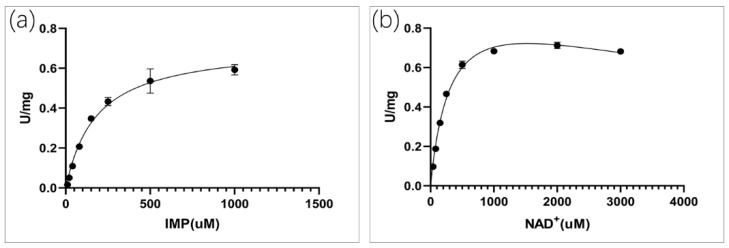
Enzyme activity of *C*Las IMPDHΔ98-201. (**a**): Varying concentrations of IMP at a fixed concentration of NAD^+^ (2 mM); (**b**): Varying concentrations of NAD^+^ at a fixed concentration of IMP (1 mM).

**Figure 5 molecules-25-02313-f005:**
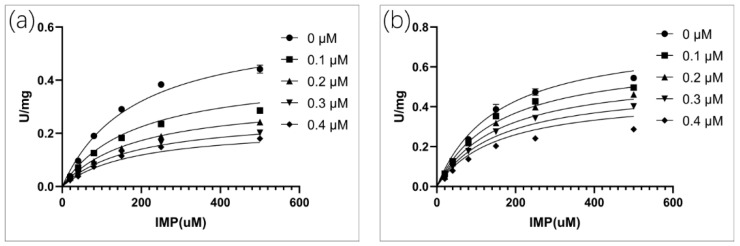
Inhibition kinetics at different concentrations of compounds by varying the IMP concentrations at a fixed NAD^+^ concentration. (**a**): Bronopol; (**b**): Disulfiram.

**Table 1 molecules-25-02313-t001:** Data collection and refinement statistics.

	*C*Las IMPDHΔ98-201
Beamline	SSRF BEAMLINE BL17U
wavelength (Å)	0.9792
Detector	ADSC QUANTUM 315r
resolution range (Å)	42.47–2.55
space group	C 1 2 1
unit cell parameters (Å)	a = 143.13, b = 134.86, c = 85.62
no. of residues/protein	390
Monomer molecular weight (kDa)	41.0
phasing method	MR
search model	chains A of 4R7J
Refinement resolution range (Å)	42.46–2.55
no. of reflections	50928
σ cutoff	1.36
R_work_	0.222
R_free_	0.264
mean B factor (Å2)	69.3
data completeness (%)	98.2
redundancy	2.58
Ramachandran plot [most favored/outliers (%)]	95.2/0.5
PDB entry	6KCF

**Table 2 molecules-25-02313-t002:** Detailed summary of the docking binding affinities (kcal/mol).

Name	Molecular Weight (g/mol)	-CDOCKER_ENERGY (kcal/mol)
Disulfiram	296.54	25.0346
Mercaptopurine	152.18	16.6785
Bronopol	199.99	11.1913
Ebselen	274.18	6.24157
Mycophenolic_acid	320.34	5.38177
Mizoribine	259.22	4.50734
Ribavirin	244.20	−5.62032
Mycophenolate_mofetil	433.50	−43.3176

**Table 3 molecules-25-02313-t003:** Inhibition of *C*Las IMPDHΔ98-201 by eight inhibitors.

Inhibitor	IMP K_i_ (µM)
Bronopol	0.23 ± 0.01
Disulfiram	0.62 ± 0.04
Ebselen	4.13 ± 0.19
Mycophenolic acid	2.43 ± 0.10
Mercaptopurine	165 ± 9.89
Mycophenolate mofetil	24.42 ± 1.65
Mizoribine	307.7
Ribavirin	>3500

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
