# Peer review of "Evaluation of Bronopol and Disulfiram as Potential Candidatus Liberibacter asiaticus Inosine 5′-Monophosphate Dehydrogenase Inhibitors by Using Molecular Docking and Enzyme Kinetic"

_molecules, 2020, doi:10.3390/molecules25102313_

Round 1
Reviewer 1 Report
This paper reports the crystal structure of Candidatus Liberibacter asiaticus IMPDH. In addition, molecular docking of eight potential inhibitors was presented with validation against the bacteria IMPDH demonstration that bronopol and disulfiram are moderately potent inhibitors (Ki values of 240 and 550 nM, respectively. Overall, I think this paper will be of interest to the scientific community focused on bacterial IMPDH inhibitors. Therefore, I recommend this paper for publication in Molecules after minor revisions as outlined below.
- In the abstract, “promising” should be changed to “potential” since the activity against the organism has not been demonstrated. It remains a possibility that the organism has salvage mechanisms that would allow it to survive IMPDH inhibition. It is not clear if rapid proliferation would necessarily mean that salvage pathways are not sufficient for all bacteria (lines 56-57). In addition, the formidable hurdle of bacteria penetration has not yet been demonstrated.
- Line 79 “Arylurea” should be “arylurea”.
- The Ki value of raibavirin could be listed as > a particular concentration since it is essentially inactive.
- Line 189, why use the abbreviation MZP? Also a Ki of 505 microM should not be characterized as potent.
Reviewer 2 Report
The paper by Nan and colleagues describes the crystallization and preliminary characterization of IMPDH from Candidatus liberibacter asiaticus, the causative agent of the citrus disease huanglonbing. The authors suggest that IMPDH is a potential target for the design of pesticides targeting this bacterium, and this work lays for the foundation for such an effort. However, several issues must be addressed.
Major issues
- Figure 2a depicts the crystallographic unit, not a native tetramer- the four subunits are from two separate biological tetramers.
- The authors screen 8 inhibitors, and find two with submicromolar Ki values. However these two compounds, bronopol and disulfiram (and also ebselen) react with thiols and are expected to inactivate the enzyme by covalently modifying the active site Cys. Therefore time dependent inhibition would probably be observed, which cannot be characterized simply with a Ki.
- The other compounds include several prodrugs of IMPDH that are not expected to inhibit in their unactivated forms. It is the nucleotide monophosphates of ribavirin, mizoribine and mercaptopurine that inhibit IMPDH, so it is not surprising that these compounds do not inhibit (or dock favorably).
- The description of the inhibition assay says the "Ki for IMP was determined"- it should say Ki for mycophenolic acid etc. Which mechanism did the data fit to? If mixed, there should be two Ki values). This section needs to include the concentration of enzyme and varied concentrations of inhibitor.
Minor issues
- Is the size on SEC consistent with a tetramer?
- Figure 3- fonts should be bigger
- line 147-148, " Because…" is not a sentence.
- How much enzyme is in an assay?
